# Microbial Risk Assessment of *Escherichia coli* O157:H7 in Beef Imported from the United States of America to Taiwan

**DOI:** 10.3390/microorganisms8050676

**Published:** 2020-05-06

**Authors:** Keng-Wen Lien, Meng-Xuan Yang, Min-Pei Ling

**Affiliations:** 1Institute of Food Science and Technology, National Taiwan University, Taipei City 10617, Taiwan; d98b47102@ntu.edu.tw; 2Department of Food Science, National Taiwan Ocean University, Keelung City 20224, Taiwan; b1abyl7on@gmail.com

**Keywords:** *Escherichia coli* O157:H7, imported beef, health risk assessment

## Abstract

Outbreaks of foodborne illness caused by pathogenic *Escherichia coli* (*E. coli*) O157:H7, which are attributable to the consumption of undercooked beef, have occurred in many countries. In Taiwan, the production of domestic beef accounts for only 5% of the total amount of beef sold. Therefore, we applied different contextual assumptions to develop a quantitative microbial risk assessment of *E. coli* O157:H7 and evaluated the risk of illness in the Taiwanese population consuming beef imported from the United States of America. The probability distribution showed that, in males aged 19–65 years in the Taiwanese population, for example, when rare beef was consumed 100 servings, there was a 90% probability of randomly intaking seven colony forming units of *E. coli* O157:H7. When medium beef was consumed 10,000 servings, there was a 90% probability of randomly intaking two colony forming units of *E. coli* O157:H7. When the exceedance risk was 5%, the rate of foodborne illnesses caused by consuming rare beef contaminated with *E. coli* O157:H7 was 10–28 cases per 1 million individuals. For medium beef, this rate was 6–13 per 100 million individuals. Sensitivity analyses indicated that the amount of *E. coli* O157:H7 remaining in beef products after cooking was the most important risk factor, followed by the amount of beef products consumed. Proper cooking of imported beef consumed by the Taiwanese public reduces the incidence of foodborne disease to almost zero, without risk of harm to health.

## 1. Introduction

Foodborne illnesses are caused by the consumption of contaminated food or drink, and common pathogenic factors include microorganisms, natural toxins, parasites, and harmful chemicals. According to the 2016 “Statistics for Etiologic Agents in Food Poisoning Cases” produced by the Taiwan Food and Drug Administration (TFDA), Ministry of Health and Welfare, the incidence of foodborne illness has increased in recent years, particularly for illnesses caused by microorganisms in food [1]. In 2017, the most common pathogenic microorganism was *Vibrio parahaemolyticus* (15 cases), followed by *Bacillus cereus* (12 cases) and *Staphylococcus aureus* (seven cases) [1]. The majority of cases occur in restaurants (301–393 cases per year), homes (59–70 cases per year), and schools (64–93 cases per year) [1]. There are also approximately 2–16 cases of pathogenic *Escherichia coli* (*E. coli*) poisoning each year in Taiwan [1]. There have been many outbreaks of foodborne illness worldwide. For example, in 2011, the consumption of raw beef contaminated with *E. coli* O111 caused four deaths in Japan [2]. In the same year, sprouts contaminated with Enterohemorrhagic *E. coli* (EHEC) O104 caused dozens of deaths in Germany [3]. The most common and toxic serotype of EHEC is *E. coli* O157:H7 [4]. Infection is characterized by severe abdominal pain with severe or bloody diarrhea [5], and in some cases, vomiting or mild fever may occur. Infections generally resolve within 7–10 days; however, 15% of patients develop hemolytic uremic syndrome [6]. Although EHEC can infect anyone, pregnant women and those with weak immune systems (i.e., young children and the elderly) are at a high risk of developing serious complications following infection [5].

The non-O157 serotype of *E. coli* is less likely to cause serious illness compared with the *E. coli* O157:H7. EHEC and other Shiga toxin-producing *E. coli* strains are transmitted through the fecal-oral route [5]. The consumption of contaminated food, such as raw or undercooked ground meat, is also a common route of infection [7]. Cross-contamination from water contaminated with feces, other food products (e.g., beef and other meat products), and during food preparation (e.g., contaminated cutting boards and kitchen utensils) may also result in infection. Sources of *E. coli* O157:H7 outbreaks have included uncooked beef, ground beef, hamburger meat, and fermented sausages [8]. Taiwan produces approximately 7000 metric tons of beef annually; however, approximately 95% of beef consumed in Taiwan is imported from abroad, mainly from the United States of America (USA) (~40%) [9]. In this study, we investigated whether the imported beef products are a potential hazard to Taiwanese consumers.

Quantitative microbial risk assessment (QMRA) is one method used to assess the risks caused by pathogenic microorganisms. This approach allows the proper management of risks, and is also used to establish international standards for food trade [10].

The purpose of this study is to establish an assessment model of exposure to *E. coli* O157:H7. The probability distribution of exposure to *E. coli* O157:H7 was integrated using the *E. coli* O157:H7 dose–response relationship to assess the exceedance risk to Taiwanese individuals following the consumption of beef imported from the USA under different situations.

## 2. Materials and Methods 

### 2.1. Hazard Identification

According to the statistics on food poisoning outbreaks from 1999 to 2017 published by the TFDA, thus far, cases of food poisoning caused by pathogenic *E. coli* have not resulted in any deaths in Taiwan and there are no cases of food poisoning caused by pathogenic *E. coli* O157:H7 [1]. Research on *E. coli* O157:H7 infections in Europe, the USA, and Canada have shown that animal food products are the main vector [11]. Undercooked ground beef caused the highest number of infections, while raw milk and poultry were also responsible for infections [12]. Aside from food products, environmental water sources are also vectors of infection [11].

Many of the foods implicated in human disease are of bovine origin, and epidemiological studies have associated the contamination of crops and water supplies with cattle [10]. An outbreak of five cases of *E. coli* O157:H7 infection that occurred in Canada in 2012 was linked to frozen beef patties [13]. *E. coli* O157:H7 have also caused outbreaks of food poisoning in the USA. For example, in 2016, the consumption of contaminated beef caused illness in 25 individuals [14].

### 2.2. Exposure Assessment

#### 2.2.1. Model Framework

This study assumed that raw beef imported from the USA may be contaminated with *E. coli* O157:H7. After arrival in Taiwan, beef is transported to steakhouses at low temperatures and stored in refrigerators prior to cooking. Beef dishes can be divided into rare, medium, and medium-well categories based on the internal temperature measured in the center after cooking [15]. By combining the amount of colony forming units (CFU) of *E. coli* O157:H7 present in beef products (Log_10_ CFU/g) with the daily consumption of beef by Taiwanese individuals (g/day), we conducted a simulation analysis of exposure to *E. coli* O157:H7 among the Taiwanese population (Log_10_ CFU). The assumptions regarding this scenario are shown in Figure 1.

#### 2.2.2. Contamination Rate and Concentration of *E. coli*

The rate of contamination (*C_cont_*
*R*) with *E. coli* O157:H7 is based on the results of the United States Department of Agriculture (USDA) raw ground beef components test for the period 1995–2012 [16]. This establishes the distribution of *E. coli* O157:H7 contamination in fresh beef imported from the USA.

Bacterial concentration was based on the number of *E. coli* O157:H7 present in 1 kg of fresh ground beef for burgers sold in Canada (Log_10_ CFU/kg). This was used to estimate the concentration of *E. coli* O157:H7 (CFU/g) in fresh beef imported from the USA (*C_cont_*
*N*) [17].

The import rate (IR) was based on data obtained from the Council of Agriculture, Executive Yuan for fresh raw beef imports (tons), and domestic raw beef production (tons). These data were used to calculate the total amount (tons) of fresh beef sold in Taiwan each year. The amount (tons) of raw beef imported from the USA between 2010 and 2014 was divided by the total amount of fresh beef sold in Taiwan each year to obtain the proportion (%) of imported beef from the USA out of the total amount of fresh beef sold in Taiwan [9].

#### 2.2.3. Consumption and Preparation

The consumption rate (CR) was calculated using daily consumption rate data obtained from the Nutrition and Health Survey in Taiwan [18]. Data sources include the Nutrition and Health Survey in Taiwan (2005–2008), the Junior High School Student Nutrition and Health Survey in Taiwan (2010 and 2011), and the Elementary School Student Nutrition and Health Survey in Taiwan (2012). For the exposure groups, we referred to the Nutrition and Health Survey and divided the population into six groups according to age and sex: males aged 13–18 years, females aged 13–18 years, males aged 19–65 years, females aged 19–65 years, males aged >65 years, and females aged >65 years. Two other groups, children aged 4–12 years and females aged 19–49 years (i.e., including only females of childbearing age) were considered, leading to a total of eight groups. The rate of beef consumption (g/day) for each sex and age group was determined by calculating the daily consumption rate (consumer-only) of beef for each identity (ID), which considered the consumption rate (g/day) for each individual as well as the time period.

Regarding the temperature of transportation, it was assumed that the fresh beef was transported to Taiwan via cold chain, and subsequently distributed at low temperatures to restaurants. Thus, the temperature conditions were under −18 °C during the transportation and storage in refrigerators in restaurants.

The heating temperature refers to consumer perceptions and the acceptability of different cooking methods for beef in the United Kingdom, defining the thickness of the beef at 2.5 cm and 110 g per serving (*S*) [15]. However, the consumption amount based on individual people at each time and some subjects might have eaten more food (exceeding two servings) and some less (below one serving). Therefore, the consumption amount is not one standard serving.

The level of cooking of beef is defined by the final temperature at the center, when cooked in a closed grill at 200 °C. Rare beef is cooked for 3 min with a 60.0 °C temperature measured in the center; medium beef is cooked for 4 min with a 70.0 °C temperature at the center; and medium-well beef is cooked for 4.5 min with a 75.0 °C temperature at the center. The decimal reduction time for beef at different levels of cooking is expressed through Equation (1):(1)$${\mathrm{Log}}_{10}\left(D\right){=\mathrm{Log}}_{10}\left(D_{0}\right)-\frac{T}{z}$$
where *D* is the time (min) required to kill 90% of the population of a particular species of bacterium at a specific heating temperature. *T* is the temperature (°C), and *Z* is the slope (°C) of the thermal death curve of heat treatment of a particular type of bacterium. In this study, the benchmark (*D*_0_) was rare beef cooked to 60.0 °C with a *D* value of 2.19 min and a *Z* value of 7.09 °C [19].

This study conservatively assumed that, when the center of the beef reached the specified temperature, the external temperature was also equal to the specified temperature. The bacteria remaining were calculated for the specified temperature for a duration of 10 s according to Equation (2):(2)$${\mathrm{Log}}_{10}\left(C\right){=\mathrm{Log}}_{10}\left(C_{cont}\;N\right)-\frac{t}{D}$$
where *C* is the residual amount of *E. coli* O157:H7 (CFU/g) in the beef after cooking under specific temperature and time conditions, Ccont N is the amount of *E. coli* O157:H7 contamination (CFU/g) in fresh beef [17], *t* is the cooking time (min), and *D* is the *D* value (min) for the specific temperature.

#### 2.2.4. Probabilistic Analysis Method

The distribution of *E. coli* O157:H7 (CFU/g) after cooking beef under different conditions was combined with the distribution of the contamination rate (%) of *E. coli* O157:H7 in fresh beef imported from the USA. This was also used to estimate the probability distribution of *E. coli* O157:H7 remaining in rare, medium, and medium-well steaks via Equation (3): (3)$$\mathrm{Dose}\;(\mathrm{CFU}/g)=\sum_{j=1}^{n}C_{ij}\times C_{cont}\;R_{j}$$
where *C**_ij_* is the distribution of *E. coli* O157:H7 remaining in food cluster *j* after cooking (Log_10_ CFU/g) refer to Equation (2), the distribution of *E. coli* O157:H7 before cook obtain from *C_cont_*
*N*. Food cluster was matched by food consumption data from foods consumed by individuals into a food group, i.e., ribeye, top cap, steak, and sirloin, etc., will pool into a beef group. The *i* is for the age and gender group. *C_cont_ R_j_* is the distribution of the *E. coli* O157:H7 contamination rate (%) in food cluster *j* obtain from USDA [16].

To establish the probability distribution of exposure to *E. coli* O157:H7 among the Taiwanese population through the consumption of beef (Log_10_ CFU), calculated according to Equation (4) shown below, we combined the following factors: (a) the amount of *E. coli* O157:H7 (CFU/g) remaining after cooking; (b) the distribution of the contamination rate (%) of *E. coli* O157:H7 in fresh beef imported from the USA; (c) the proportion of beef imported from the USA out of the total amount of fresh beef sold in Taiwan (%); and (d) the distribution of the consumption rate (g/day) of beef products for the different sex and age groups.
(4)$${P(D}_{ij})=\sum_{j=1}^{n}{\mathrm{Dose}\times CR}_{ij}\times {IR}_{j}$$
where *P*(*D_ij_*) is the probability distribution of exposure to *E. coli* O157:H7 (Log_10_ CFU) of individual for each sex and age exposure group via the consumption of food cluster *j*. *CR_ij_* is the distribution of the consumption rate (g/day) when consuming food cluster *j* for each individual ID *i* in the exposure group. Finally, *IR_j_* is the distribution of the proportion of beef imported from the USA out of the total amount of fresh beef sold in Taiwan (%). Equation (4) establishes the probability distribution of ingesting *E. coli* O157:H7 through beef products among the Taiwanese population for each sex and age group (Log_10_ CFU).

### 2.3. Dose Response

We referred to the modified beta-binomial model based on the beta-Poisson model to establish the dose–response relationship for the amount of *E. coli* O157:H7 ingested (Log_10_ CFU) and the incidence of foodborne illness (%). The dose–response relationship chart refers to that reported by Cassin et al., 1998; the parameter α is 0.267 and ln β is the normal distribution (5.435, 2.47) [15].

### 2.4. Risk Characterization

We used a probability analysis and the concept of exceedance risk to assess the risk of exposure to *E. coli* O157:H7 from eating beef products in the Taiwanese public in different sex and age groups. By combining the probability analysis for exposure and the dose–response relationship, we could estimate the cumulative risk to quantify the exceedance risk for the incidence of foodborne diseases caused by ingesting *E. coli* O157:H7 when consuming beef products.

The exceedance risk is based on the cumulative probability risk and calculates the amount of risk produced under the conditional probability in a particular situation and the probability of exceeding this risk state. The risk probability results can be used to create the cumulative risk curve. Points on the horizontal axis represent the risk of the incidence of foodborne illness. The vertical axis represents the potential exceedance risk and can be expressed by Equation (5):(5)$$P(E_{D})=P(D)\times P(R/D)$$
where *P*(*E_D_*) is the exceedance risk of the incidence of a specific foodborne illness, *P*(*D*) is the distribution of ingesting *E. coli* O157:H7 in beef products among the Taiwanese population for each sex and age group, and *P*(*R/D*) is the dose–response relationship from ingesting *E. coli* O157:H7. 

When performing risk assessments, there is often insufficient information to describe the actual exposure, increasing the uncertainty of the risk assessment and resulting in a large deviation from the actual situation. Therefore, after risk quantification, it is necessary to provide an explanation of any uncertainties, including assumptions, integration, and the basis for appraisals.

This study used the sensitivity analysis included in the statistical software Crystal Ball (Version 5.2.2; Decisioneering Inc., Denver, CO, USA) to compare the distribution of *E. coli* O157:H7 after cooking beef products, all parameters show in Table A1. The distribution of the contamination rate (%) of *E. coli* O7H7 in fresh beef imported from the USA, the distribution of the consumption rate (g/day) of beef products for different sex and age groups, and the distribution of the proportion of beef imported from the USA (%). This analysis was conducted to establish the influence of these four distribution parameters on ingesting *E. coli* O157:H7 from consuming beef products among the Taiwanese population for each sex and age group (Log_10_ CFU).

## 3. Results

### 3.1. Distribution of Variables

The distribution of the contamination rate (%) of *E. coli* O157:H7 in fresh beef imported from the USA was log normal (LN) (0.217%, 2.38). The distribution of the concentration of *E. coli* O157:H7 was also LN (0.02 CFU/g, 18.19). The distribution of the proportion of beef imported from the USA out of the total amount of fresh beef sold in Taiwan (including beef imported from other countries and locally produced beef) was normal (N) (30%, 0.06). The distribution of the daily consumption rate of beef for each sex and age group is shown in Table 1.

We calculated the D value for *E. coli* O157:H7 at different temperatures using Equation (1) for rare (60.0 °C), medium (70.0 °C), and medium-well (75.0 °C) beef: 2.19 min, 8.51 × 10^−2^ min, and 1.68 × 10^−2^ min, respectively. We also calculated the distribution of *E. coli* O157:H7 remaining in each meal after cooking using Equation (2) and obtained the following values: LN (2.09 × 10^−2^ CFU/g, 18.19), LN (2.75 × 10^−4^ CFU/g, 18.19), and LN (2.91 × 10^−12^ CFU/g, 18.2), respectively. For beef imported from the USA, the distribution of *E. coli* O157:H7 (CFU/g) remaining in beef after cooking was combined with the distribution of the contamination rate (%) of *E. coli* O157:H7, as shown in Figure 2.

### 3.2. Intake of E. coli O157:H7

The probability distribution of *E. coli* O157:H7 intake (Log CFU) from consuming beef imported from the USA in the Taiwanese population is shown in Table 2.

The probability distribution of ingesting *E. coli* O157:H7 among the Taiwanese population for each sex and age group from consuming rare beef imported from the USA shows that each of the following groups has a 90% maximum likelihood of ingesting a specific number of *E. coli* O157:H7 bacteria on one occasion for every 100 instances of rare beef consumption: (1) individuals aged 4–12 years, four *E. coli* O157:H7 bacteria; (2) males aged 13–18 years, two *E. coli* O157:H7 bacteria; (3) females aged 13–18 years, one *E. coli* O157:H7 bacterium; (4) males aged 19–65 years, seven *E. coli* O157:H7 bacteria; (5) females aged 19–65, six *E. coli* O157:H7 bacteria; (6) males aged >65 years, two *E. coli* O157:H7 bacteria; (7) females aged >65 years, two *E. coli* O157:H7 bacteria; and (8) females aged 19–49 years (of childbearing age), one *E. coli* O157:H7 bacterium (Figure A1).

The probability distribution of ingesting *E. coli* O157:H7 among the Taiwanese population for each sex and age group from consuming medium beef imported from the USA shows that each of the following groups has a 90% maximum likelihood of ingesting a specific number of *E. coli* O157:H7 bacteria on one occasion for every 10,000 instances of medium beef consumption: (1) individuals aged 4–12 years, three *E. coli* O157:H7 bacteria; (2) males aged 13–18 years, four *E. coli* O157:H7 bacteria; (3) females aged 13–18 years, two *E. coli* O157:H7 bacteria; (4) males aged 19–65 years, two *E. coli* O157:H7 bacteria; (5) females aged 19–65 years, four *E. coli* O157:H7 bacteria; (6) males aged >65 years, two *E. coli* O157:H7 bacteria; (7) females aged >65 years, one *E. coli* O157:H7 bacterium; and (8) females aged 19–49 years (of childbearing age), four *E. coli* O157:H7 bacteria (Figure A2).

The probability distribution of ingesting *E. coli* O157:H7 among the Taiwanese population for each sex and age group from consuming medium-well beef imported from the USA shows that each of the following groups has a 90% maximum likelihood of ingesting a specific number of *E. coli* O157:H7 bacteria on one occasion for every 10^12^ instances of medium-well done beef consumption: (1) individuals aged 4–12 years, four *E. coli* O157:H7 bacteria; (2) males aged 13–18 years, five E. coli O157:H7 bacteria; (3) females aged 13–18 years, two *E. coli* O157:H7 bacteria; (4) males aged 19–65 years, three *E. coli* O157:H7 bacteria; (5) females aged 19–65 years, four E. coli O157:H7 bacteria; (6) males aged >65 years, two *E. coli* O157:H7 bacteria; (7) females aged >65 years, two *E. coli* O157:H7 bacteria; and (8) females aged 19–49 years (of childbearing age), two *E. coli* O157:H7 bacteria (Figure A3).

### 3.3. Exceedance Risk

The probability distributions of ingesting *E. coli* O157:H7 among the Taiwanese population for each sex and age group from consuming rare beef imported from the USA was compared against the dose–response relationship to obtain the incidence of foodborne illness under different situations. The accumulated risk for the incidence of foodborne illness was used to quantify the exceedance risk among the Taiwanese population for the incidence of foodborne diseases caused by ingesting *E. coli* O157:H7 when consuming beef products (Equation (5)). The exceedance risks are shown in Figure 3.

When the risk was 5%, the results showed that the incidence of foodborne illness from consuming rare and medium beef among 1 million and 100 million individuals, respectively, in each group was as follows: (1) individuals aged 4–12 years, 14 and nine; (2) males aged 13–18 years, 27 and 10; (3) females aged 13–18 years, 23 and eight; (4) males aged 19–65 years, 28 and 13; (5) females aged 19–65 years, 10 and eight; (6) males aged >65 years, 13 and six; (7) females aged >65 years, 18 and eight; and (8) females aged 19–49 years (of childbearing age), 21 and 10.

### 3.4. Sensitivity Analysis

The rank correlation results of the sensitivity analysis are shown in Figure 4. The highest correlation was estimated dose, obtained for the amount of *E. coli* O157:H7 bacteria remaining in the beef after cooking, followed by a contamination rate of beef with *E. coli* O157:H7, a consumption rate, and imported ratio.

## 4. Discussion

### 4.1. Quantitative Microbial Risk Assessment (QMRA) for E. coli O157:H7

In recent years, many countries have applied QMRA to assess the risk of foodborne illnesses in individuals consuming contaminated foods. The results assist governments in developing management policies to improve food safety. However, there is currently no comprehensive QMRA research conducted for specific groups and foods in Taiwan. Owing to differences in the dietary habits of the Taiwanese population, varied results may be observed in comparison to the risk assessments conducted in other countries for foodborne illnesses caused by ingesting *E. coli* O157:H7 in beef products.

Comparatively, Kiermeier et al. investigated the incidence of foodborne illness from consuming hamburgers made from imported Australian beef in fast food restaurants in the USA (2.2 × 10^−6^–1.5 × 10^−5^) [6]. This rate was similar to our observed in Taiwanese males aged 19–65 years from consuming rare beef (i.e., 2.8 × 10^−5^) with an exceedance risk of 0.05. On the other hand, the incidence of foodborne illness from consuming medium beef with an exceedance risk of 0.05 in males aged 19–65 years was 1.3 × 10^−7^, which is also lower than that which Kiermeier et al. reported [6]. In practice, when importing the beef from the USA, contaminated beef should be removed from the supply chain after microbiological testing in import or export inspection; thus, it is expected that this incidence is lower. Moreover, there were no *E. coli* O157:H7 contamination data for intact beef or steak. Therefore, we referred to the contamination data published for ground beef. Although this tends to overestimate the bacterial contamination of real beef cuts, it is rational to use the data for a conservative estimate in risk assessment.

We use exceedance risk for knowing the possibilities that may occur in different specific morbidity scenarios. When the exceedance risk was 0.05, the following results were observed: (1) The incidence of foodborne illness from consuming rare beef in males aged 19–65 years was 2.8 × 10^−5^. Therefore, the incidence rate of foodborne illness was >1 in 10,000 individuals and the probability of occurrence was 5%. (2) The incidence of foodborne illness from consuming medium beef in males aged 19–65 years was 1.3 × 10^−7^. Thus, the incidence rate of foodborne illness was >2 in 1 million individuals and the probability of occurrence was 5%. (3) The incidence of foodborne illness from consuming rare beef in females aged 19–65 years was 9.3 × 10^−6^. Hence, the incidence rate of foodborne illness was >10 in 10,000 individuals and the probability of occurrence was 5%. (4) The incidence of foodborne illness from consuming medium beef in females aged 19–65 years was 7.8 × 10^−8^. Thus, the incidence rate of foodborne illness was >8 in 10 million individuals and the probability of occurrence was 5%.

Comparison of the risk calculated in the present study with that reported by Cassin et al., only when the exceedance risk was 0.05 that the incidence of foodborne illness from consuming rare and medium beef in the Taiwanese population is higher than that recorded in the Canadian population caused by ingesting *E. coli* O157:H7 in beef patties [17]. In other words, when consuming medium beef in the Taiwanese population had a 95% chance, the incidence of foodborne illness for every 100 million serving beef contaminated with *E. coli* O157:H7 was under 1.3 × 10^−8^. Our results were also lower than risk assessments annual illnesses from ground beef consumption in the United States (1.0 × 10^−6^) [20]. Moreover, the incidence of foodborne illness from consuming medium-well beef (near zero) was lower than that noted among the Canadian population.

In present report, under the most conservative scenario, there was approximately a 5% probability that the incidence of foodborne illness from consuming rare steak would be >2.8 × 10^−5^ (95% probability will under 2.8 × 10^−5^) and the highest incidence of foodborne illness was 1.43%; however, the probability of occurrence was only 0.01%, in other words, 99.9% of illness will be under 1.43%. There was approximately a 5% probability that the incidence of foodborne illness from consuming medium beef would be >1.3 × 10^−7^. The highest incidence of foodborne illness was 8.21 × 10^−6^; however, the probability of occurrence was only 0.01%.

Cassin et al. studied the risk of foodborne illness caused by ingesting *E. coli* O157:H7 in beef patties in Canada [17]. Their results showed that the incidence of foodborne illness from consuming one beef patty was 1.0 × 10^−22^–1.0 × 10^−2^. The average incidence of foodborne illness for each beef patty in adult and child populations was 5.1 × 10^−5^ and 3.7 × 10^−5^, respectively. The sensitivity analysis showed that the amount of *E. coli* O157:H7 bacteria in bovine feces exhibited the highest correlation (~0.6), followed by host susceptibility (~0.6), and carcass contamination (~0.3). In our study, the highest correlation was estimated dose (~0.909), followed by a contamination rate of beef with *E. coli* O157:H7 (~0.307), and consumption rate (~0.263). The estimated dose of *E. coli* O157:H7 bacteria had the most impact on the mean number of illnesses in our model.

### 4.2. Uncertainty

In our study, the scenario uncertainty could be caused by the following conditions (1) Assessment of beef imported from the USA only: the domestic production of beef accounts for only 5% of the total beef sold, and *E. coli* O157:H7 has not been found in domestic beef products. USA beef accounts for the largest proportion of the imported beef (approximately 30%). Therefore, only beef products made from beef imported from the USA were assessed. (2) Transportation temperature and time: under actual conditions, beef may be refrigerated when imported and the transportation times may vary. However, the present study assumed that changes in the number of *E. coli* O157:H7 in raw beef were not significant, regardless of whether it was frozen or refrigerated during transportation. (3) Cooking conditions: the assumptions of this study regarding the thickness, weight, and cooking temperature and time for each piece of beef may differ from the actual conditions in restaurants. (4) Cross-contamination: the risk of cross-contamination after cooking beef was not considered and may be underestimated. (5) Consumption rate: the present study only calculated individuals who ate beef (consumer-only) and excluded a consumption rate = 0. Therefore, it is not to assess the actual food intake in the whole population. In addition, Taiwanese individuals do not consume beef frequently which may have resulted in an overestimation of the consumption rate. The present study assumed that the daily consumption rate was the consumption rate for each incidence. However, this may differ from the actual daily consumption of beef.

The parameter uncertainties contained in the present study included the following aspects: (1) The number of contaminated bacteria: the amount of *E. coli* O157:H7 contamination was based on the research hypotheses proposed by Cassin et al. [17] rather than the actual test results for *E. coli* O157:H7 in fresh beef sold in the USA or Taiwan. (2) Contamination rate: the contamination rate of *E. coli* O157:H7 is derived from domestic inspections by the USDA rather than inspections of fresh beef imported into Taiwan from the USA. (3) Dose–response relationship: in the present study, the dose–response relationship was based on the adult population [17]. The actual dose–response relationship may differ among individuals with low immunity, such as children and the elderly.

## 5. Conclusions

We want to emphasize again, thus far, that there are no cases of food poisoning caused by pathogenic *E. coli* O157:H7 and also have not resulted in any deaths in Taiwan. The present article reports the results of a risk assessment of *E. coli* O157:H7 from the consumption of beef steak made and imported only from USA manufacturing beef. Starting with the prevalence and concentration of *E. coli* O157:H7 in USA beef, from transportation through to cooking and consumption in Taiwan. Based on the results of the present study, we recommend the following: (1) importers should strictly control and manage testing for pathogenic bacteria to reduce contamination by bacteria; (2) the public should reduce their consumption of high-risk beef products, such as rare beef; and (3) proper cooking of beef imported from the USA reduces the incidence of foodborne disease in Taiwanese individuals to almost zero, without risk of harm to health.

## Figures and Tables

**Figure 1 microorganisms-08-00676-f001:**
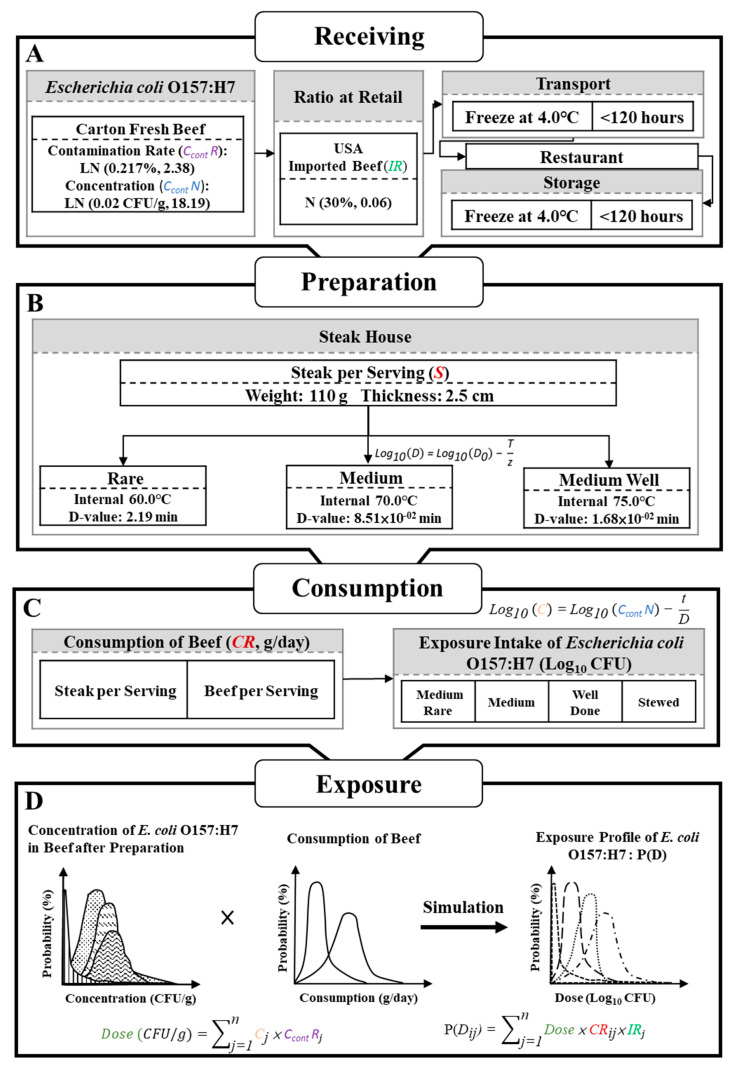
Assumptions for various scenarios of consumption of beef imported from the United States of America to Taiwan when beef is infected by *Escherichia coli* O157:H7. (**A**) Receiving, transportation, and storage of beef, (**B**) Preparation of beef, (**C**) Consumption of beef, and (**D**) Exposure profile of E. coli O157:H7. *Note:* LN = log normal distribution; N = normal distribution; D-value = decimal reduction time; CFU = colony forming units.

**Figure 2 microorganisms-08-00676-f002:**
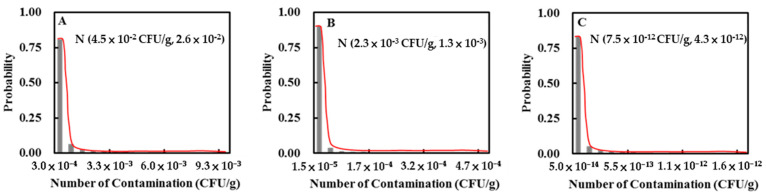
Probability distribution of *E. coli* O157:H7 contamination in beef imported from the United States of America after cooking to (**A**) rare, (**B**) medium, and (**C**) medium-well. Note: CFU = colony forming units.

**Figure 3 microorganisms-08-00676-f003:**
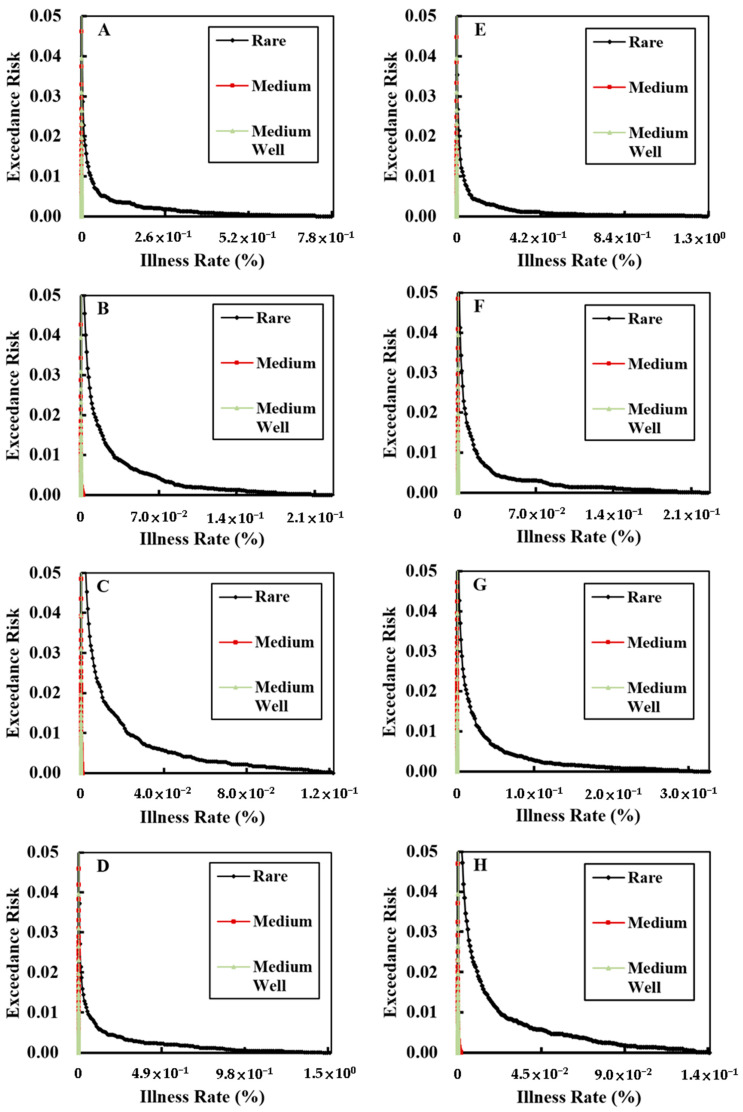
Exceedance risk to the Taiwanese population from consuming beef imported from the United States of America. (**A**) Males and females aged 4–12 years, (**B**) males aged 13–18 years, (**C**) females aged 13–18 years, (**D**) males aged 19–65 years, (**E**) females aged 19–65 years, (**F**) males aged >65 years, (**G**) females aged >65 years, and (**H**) females aged 19–49 years (of childbearing age).

**Figure 4 microorganisms-08-00676-f004:**
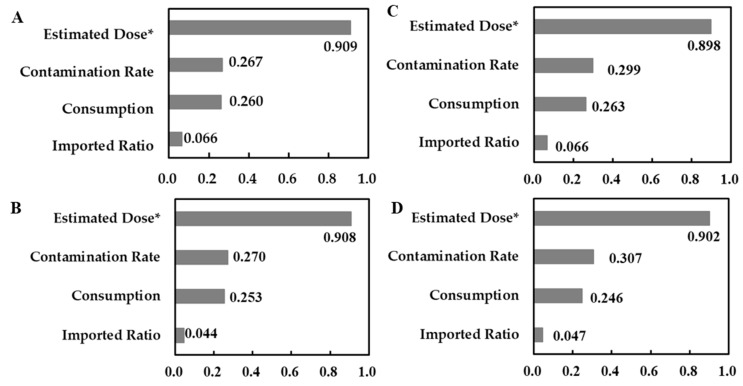
Rank correlation results of the sensitivity analysis for the consumption of rare and medium beef by males and females aged 19–65 years. (**A**) Males consuming rare beef, (**B**) males consuming medium beef, (**C**) females consuming rare beef, and (**D**) females consuming medium beef. * Unit: colony forming units.

**Table 1 microorganisms-08-00676-t001:** Distribution of the daily consumption rate of beef for each sex and age group.

Age Group	Sex	Daily Consumption (g/day)
4–12	Male and female	LN ^1^ (52.69, 2.60)
13–18	Male	LN (78.34, 2.10)
13–18	Female	LN (61.32, 2.24)
19–65	Male	LN (74.53, 2.41)
19–65	Female	LN (53.08, 2.75)
>65	Male	LN (40.14, 2.21)
>65	Female	LN (70.44, 1.33)
19–49	Female	LN (54.71, 2.81)

^1^ LN = log normal distribution.

**Table 2 microorganisms-08-00676-t002:** Probability distribution of *E. coli* O157:H7 intake (Log CFU) from consuming beef imported from the United States of America in the Taiwanese population.

Group	Intake (Log CFU ^1^)	Group	Intake (Log CFU)
Males aged 13–18 years		Males aged >65 years	
Rare	N ^b^ (−1.74, 0.24)	Rare	N (−1.78, 0.18)
Medium	N (−3.35, 0.18)	Medium	N (−3.77, 0.18)
Medium–well	N (−11.27, 0.14)	Medium–well	N (−11.86, 0.23)
Females aged 13–18 years		Females aged >65 years	
Rare	N (−1.94, 0.26)	Rare	N (−1.65, 0.18)
Medium	N (−3.75, 0.21)	Medium	N (−4.05, 0.31)
Medium–well	N (−11.83, 0.27)	Medium–well	N (−11.71, 0.21)
Males aged 19–65 years		Males and females aged 4–12 years	
Rare	N (−1.15, 0.13)	Rare	N (−1.37, 0.13)
Medium	N (−3.67, 0.21)	Medium	N (−3.49, 0.14)
Medium–well	N (−11.52, 0.18)	Medium–well	N (−11.37, 0.14)
Females aged 19–65 years		Females aged 19–49 years (of childbearing age)	
Rare	N (−1.21, 0.13)	Rare	N (−1.92, 0.24)
Medium	N (−3.44, 0.17)	Medium	N (−3.45, 0.14)
Medium–well	N (−11.36, 0.16)	Medium–well	N (−11.79, 0.24)

^1^ N = normal distribution; ^b^ CFU = colony forming units.

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
