# Peer review of "Microbial Risk Assessment of Escherichia coli O157:H7 in Beef Imported from the United States of America to Taiwan"

_microorganisms, 2020, doi:10.3390/microorganisms8050676_

Round 1

Reviewer 1 Report

THe study deals with QMRA associated with E. coli O157 H7 in beef products imported from USA. The paper is well-written. However there may be some improvements in the description of the model. There are some equations but all variables are not provided with their respective distributions. A recap table would be useful and would bring rigour to the analysis and modeling part.  There are also some parts which sounds like catalog. I mean that sometimes some parts are too long and there is no real opinion of the author. The author should sometimes analyse the results and should take a position. The reader is indeed a bit lost among a big amount of figures. The role of the author is to guide the reader towards the conclusions that he thinks relevant.

Here are my comments throughout the document:

It would be instructive to have the symbols used in equations in figure 1 or a table providing parameters and equations, distributions and references.

Lines 103 and 156 distribution of E. coli O157 H7 in imported beef from USA should be given as a symbol and included in equations. The whole sequence of equations should be provided. Here it begins with remaining E. coli after cooking.

Line 160 what index i refers to ? what is food cluster ?

Lines 164 – 168 : there may be some confusion between figures under brackets and equations.

Line 174 : individual ID i. Does it refer to sex and gender class ? Not clear. If not, where does it taken into account in equations ?

Line 199 : does the dose response relationship take into account the different sensitivities of people like children for example. Which dose response was used ? there is no reference.

Line 238 : in table 1 I do not see the distributions for children aged between 4 and 12.

Lines 243 – 249 : please clarify. Especially what does mean : on one occasion for every 10,000 instances of medium beef consumption ?

Lines 250-259 Results should be given in a table.

Lines 290-300 : It’s not necessary to list all figures since the tendencies are similar. Results should be summarized by giving tendencies and ranges.

Discussion from line 316 : not clear which figures refer to the authors’ study and those which are cited.

Line 339 ; no hypothesis is given to explain why incidence is higher in the Taiwanese population.

The uncertaintly section is interesting and exhaustive but it should be shorten. Are they classified by importance order ? If not it should.

Regarding conclusions, is it possible to make a comparison with real incidence of foodborne diseases ?

There is no mention of children who are the most susceptible to E. coli O157 H7.

Author Response

Responses to Reviewer 1 Comments (Manuscript ID: microorganisms-763573)

We thank the reviewer’ valuable comments. These comments greatly improve the quality of this manuscript. The manuscript has been revised with the comments carefully considered. We used the blue markings to highlight the changes and new entries in the text, respectively. The following are the detailed responses to each comment.

1. It would be instructive to have the symbols used in equations in figure 1 or a table providing parameters and equations, distributions and references.

We have added the summary of parameters in Appendix A Table 1A.

2. Lines 103 and 156 distribution of E. coli O157 H7 in imported beef from USA should be given as a symbol and included in equations. The whole sequence of equations should be provided. Here it begins with remaining E. coli after cooking.

We have revised this section (L100-142).

3. Line 160 what index i refers to ? what is food cluster ?

We have revised this section. The i is for the age/gender group.  But under this model, the age/gender does not affect level, i.e. there is no indication that >65 group prefers rare cooking. So C should only have a j subscript.

Food cluster was matched by food consumption data from foods consumed by individuals into a food group.

4. Lines 164 – 168 : there may be some confusion between figures under brackets and equations.

We have revised brackets in this section (L169-173).

5. Line 174 : individual ID i. Does it refer to sex and gender class ? Not clear. If not, where does it taken into account in equations ?

Yes, the i is for the age/gender group. We have revised this sentence.

6. Line 199 : does the dose response relationship take into account the different sensitivities of people like children for example. Which dose response was used ? there is no reference.

In the present study, the dose–response relationship was based on the adult population. Actually, dose–response relationship may differ among individuals with low immunity, such as children and the elderly.

However, there were no children or elderly dose-response data that could use. Therefore, we referred to the adult data to assess the exceedance risk in Fig 3 (A). This may underestimate the risk; we mention this in the uncertainty.

7. Line 238 : in table 1 I do not see the distributions for children aged between 4 and 12.

We have shown the males and females aged 4–12 years group in table 1 and 2. We do not have separate into two groups because they are not sexually mature males and females.

8. Lines 243 – 249 : please clarify. Especially what does mean : on one occasion for every 10,000 instances of medium beef consumption?

Our study only refers to the incidence rate after one instance of consumption. Taking the data in Table 2 as an example (the calculation was made assuming 1,000,000 instances of consumption), men aged 19 to 65 years had a sum total intake of 7.1*10^4 CFU E. coli O157:H7 when consuming rare steak.

9. Lines 250-259 Results should be given in a table.

The results could be present in figure, we added the Figure A1-A3 in the Appendix A.

10. Lines 290-300 : It’s not necessary to list all figures since the tendencies are similar. Results should be summarized by giving tendencies and ranges.

We have revised and shorten this section (L295-298).

11. Discussion from line 316 : not clear which figures refer to the authors’ study and those which are cited.

We have revised this section (L314-362).

12. Line 339 ; no hypothesis is given to explain why incidence is higher in the Taiwanese population.

Only when exceedance risk was 0.05. We have revised this section.

13. The uncertaintly section is interesting and exhaustive but it should be shorten. Are they classified by importance order ? If not it should.

We have revised and shorten this section (L364-389).

14. Regarding conclusions, is it possible to make a comparison with real incidence of foodborne diseases?

Thus far, there are no cases of food poisoning caused by pathogenic E. coli O157:H7 and also have not resulted in any deaths in Taiwan. So it’s not possible to make a comparison with real incidence of foodborne diseases in Taiwan. To avoid misunderstanding, we have revised this section.

15. There is no mention of children who are the most susceptible to E. coli O157 H7.

Dose–response relationship: in the present study, the dose–response relationship was based on the adult population. But actual dose–response relationship may differ among individuals with low immunity, such as children and the elderly.

However, there were no children or elderly dose-response data. Therefore, we referred to the adult data to assess the exceedance risk in Fig 3. This may underestimate the risk, we mention this in the uncertainty.

Reviewer 2 Report

L13 Outbreak is usually reserved for disease. Do you mean outbreaks of foodborne illness?

L19-20 Is consuming rare beef 100 times a realistic exposure scenario? Also, what is the unit of time, 100 consumptions per year?

L24 10-28 cases per million per year? Need unit time for all case estimates

Figure 1 - Recommend denoting each variable in Figure 1 with the same symbology used in the equations so it is possible to easily understand data flows through the model (e.g. Contamination rate is Rj in equation (3))

L99 Please justify the use of ground beef contamination rate and density but log reduction model based steak exposure... Processing of streak vs. ground beef is different; recommend revising cooking reduction model revision to ground beef dimensions/parameters

L160 Specificy that Cij is calculated via equation (2)

L162 Recommened refering to this "exposure" as dose for consistency with QMRA literature/framework

L170 same comment as L162, recommend using dose as exposure assessment determines the dose used in the dose-response model

L179 Revise subheading to "Dose Response"

L181 Here the authors refer to dose in units of Log10 CFU; whereas, in the dose-response relationship described by Cassin et al. the dose is in units of CFU; please clarify the input units for the dose-response model that were used

L244-245 & L253-254 & L264-265 It seems counter intuitive that younger age (4-12) would be ingest more E. coli than aged 13-18? Are age 4 to 12 ingesting more beef daily than adults? That does not seem likely.

L282-287 How does these estimated risks compare to observed cases of illness (L40-41) in comparable units of population and time.

L301 Figure 4 - Recommend changing label from "contamination" to "estimated dose" to better reflect the origin of the variable

L393-400 Same comment as L282-287

L358 Most of the exposure assessment data are from ground beef or hamburger and yet the count reduction model is based on parameters from steak, which is not conservative. Please justify use of steak parameters (dimensions) rather than ground beef in cooking reduction.

L364 "Taiwanese individuals do not consume beef frequently" - Based on this assertion is the exposure described in L19-20; 100 rare beef consumption events per year reasonable?

Author Response

Responses to Reviewer 2 Comments (Manuscript ID: microorganisms-763573)

We thank the reviewer’ valuable comments. These comments greatly improve the quality of this manuscript. The manuscript has been revised with the comments carefully considered. We used the blue markings to highlight the changes and new entries in the text, respectively. The following are the detailed responses to each comment.

1. L13 Outbreak is usually reserved for disease. Do you mean outbreaks of foodborne illness?

Yes, we have revised this sentence in L13.

2. L19-20 Is consuming rare beef 100 times a realistic exposure scenario? Also, what is the unit of time, 100 consumptions per year?

The phrase “consuming 100 times” means 100 meals or servings, not 100 consumptions per year. We have revised 100 times to 100 servings. But in this study calculated the consumption amount based on individual people at each time and some subjects might have eaten more food (exceeding two servings) and some less (below one serving). Therefore, the consumption amount is not one standard serving.

3. L24 10-28 cases per million per year? Need unit time for all case estimates.

Not for per year, we use exceedance risk for knowing the possibilities that may occur in different specific morbidity scenarios. For instance, when consuming rare steaks, men aged 19–65 years had a 99% chance to ingest 0–1 CFU E. coli O157:H7, or 1% chance to ingest over 1 CFU E. coli O157:H7. The corresponding incidence rate for an intake of 1 CFU E. coli O157:H7 was 0.034%, meaning that there was a 99% chance that the incidence rate would be 0–0.034%, and a 1% chance that the incidence rate would exceed 0.034%. As a result, 1% is the probability of exceeding a 0.034% incidence rate of the disease and the exceeding risk under such condition is 1%.

4. Figure 1 - Recommend denoting each variable in Figure 1 with the same symbology used in the equations so it is possible to easily understand data flows through the model (e.g. Contamination rate is Rj in equation (3))

We try to show more detail in fig1, but it's hard to input all information in fig 1. For easily understand data, we also have added the summary of parameters in Appendix A Table 1A.

5. L99 Please justify the use of ground beef contamination rate and density but log reduction model based steak exposure... Processing of streak vs. ground beef is different; recommend revising cooking reduction model revision to ground beef dimensions/parameters.

Indeed, the processing of streak vs. ground beef is different. This study assumed that all steaks were cuts of real beef. In the steak consumption data and cooking conditions referred to by this study all used real beef cuts. However, there were no E. coli O157:H7 contamination data for intact beef or steak. Therefore, we referred to the contamination data published for ground beef. Although this tends to overestimate the bacterial contamination of real beef cuts, it is rational to use the data for a conservative estimate in risk assessment.

6. L160 Specificy that Cij is calculated via equation (2).

Yes, we have revised this sentence in L160 to L162.

7. L162 Recommened refering to this "exposure" as dose for consistency with QMRA literature/framework.

We have revised exposure as dose.

8. L170 same comment as L162, recommend using dose as exposure assessment determines the dose used in the dose-response model.

We have revised exposure as dose.

9. L179 Revise subheading to "Dose Response"

We have revised subheading to "Dose Response".

10. L181 Here the authors refer to dose in units of Log10 CFU; whereas, in the dose-response relationship described by Cassin et al. the dose is in units of CFU; please clarify the input units for the dose-response model that were used.

In the dose-response relationship described by Cassin et al (1998), the distribution of the ingested does in terms of CFU in 3.2 but in Beta-Binomial dose–response model (fig 2) show the ingested dose is in units of Log10 CFU. In the dose-response should input the units of Log10 CFU.

11. L244-245 & L253-254 & L264-265 It seems counter intuitive that younger age (4-12) would be ingest more E. coli than aged 13-18? Are age 4 to 12 ingesting more beef daily than adults? That does not seem likely.

Indeed, that does not seem reasonable. But the consumption rate was calculated using daily consumption rate data obtained from the Nutrition and Health Survey in Taiwan. The Nutrition and Health Survey in Taiwan database is based on reviewing the diet over a 24-h period. Individuals may have forgotten or incorrectly reported their food intake. In table 1, the daily consumption rate of beef (g/day) for aged 4-12 group even more than 65 male. This is because of the “consumption rate”, the present study only calculated individuals who ate beef (consumer-only) and excluded a consumption rate = 0. Therefore, it is impossible to assess the actual food intake in the whole population, we show this in our uncertainty. In my opinion, this seems likely. Older Taiwanese individuals do not consume beef frequently, because they grow up in agricultural society, the members of which vow never to eat beef. So that younger age would be ingesting more beef and E. coli.

12. L282-287 How does these estimated risks compare to observed cases of illness (L40-41) in comparable units of population and time. L393-400 Same comment as L282-287.

Thus far, there are no cases of food poisoning caused by pathogenic E. coli O157:H7 and also have not resulted in any deaths in Taiwan. So it’s not possible to make a comparison with real incidence of foodborne diseases in Taiwan. To avoid misunderstanding, we have revised the conclusions. If to compare with the incidence of foodborne illness from consuming hamburgers in the USA or Canada, our results lower than that of their report.

13. L301 Figure 4 - Recommend changing label from "contamination" to "estimated dose" to better reflect the origin of the variable.

We have revised the Figure 4.

14. L358 Most of the exposure assessment data are from ground beef or hamburger and yet the count reduction model is based on parameters from steak, which is not conservative. Please justify use of steak parameters (dimensions) rather than ground beef in cooking reduction.

This study assumed that all steaks were cuts of real beef. In the steak consumption data and cooking conditions referred to by this study all used real beef cuts. However, there were no E. coli O157:H7 contamination data for intact beef or steak. Therefore, we referred to the contamination data published for ground beef. Although this tends to overestimate the bacterial contamination of real beef cuts, it is rational to use the data for a conservative estimate in risk assessment.

15. L364 "Taiwanese individuals do not consume beef frequently" - Based on this assertion is the exposure described in L19-20; 100 rare beef consumption events per year reasonable?

Our study only refers to the incidence rate after one instance of consumption, not a per annum type risk. Taking the data in Table 2 as an example (the calculation was made assuming 1,000,000 instances of consumption), men aged 19 to 65 years had a sum total intake of 7.1*10^4 CFU E. coli O157:H7 when consuming rare steak.

According to the Taiwan food industry research & development survey, when Taiwanese consumption meat and its product, about 70% are pork and 10% are beef, meaning that there are about 73~108 times per year will eat beef or its product. So 100 times beef consumption events per year are reasonable. But in general, Taiwanese normally not likely to eat rare beef.

Round 2

Reviewer 1 Report

The authors answered some questions. The model framework still needs to be improved. Usually in such model, all data should be retrieved from equations, even if equations seem to be straightforward. The principle of such model frameworks is that outputs from equation 1 become input for equation 2, and so on… For example, contamination rate and concentration of contaminated products are given (if I understood well). It seems that some outputs are not well described. For example, probability of consuming imported products, probability of consuming contaminated products if imported, distribution of E. coli O157 :H7 per serving before cooking, and then after cooking.

Cj and the notion of food cluster is not sufficiently described. I read your answer and I think that gender and age should be mentioned in the manuscript. What is a food group ? If there is a distinction between consumption profile, on which basis is performed this distinction ? Why is this distribution applied to already cooked products and not uncooked ?

Line 376 : was it not possible to have the distribution of the population who eat beef in Taiwan ? Indeed it is an important limitation of the study.

Author Response

The authors answered some questions. The model framework still needs to be improved. Usually in such model, all data should be retrieved from equations, even if equations seem to be straightforward. The principle of such model frameworks is that outputs from equation 1 become input for equation 2, and so on… For example, contamination rate and concentration of contaminated products are given (if I understood well). It seems that some outputs are not well described. For example, probability of consuming imported products, probability of consuming contaminated products if imported, distribution of E. coli O157 :H7 per serving before cooking, and then after cooking.

Answer:
About the probability of consuming imported products, in present study, we focus on beef Imported from the USA to Taiwan, we use beef Imported rate (IR), the amount of raw beef imported from the USA between 2010 and 2014 was divided by the total amount of fresh beef sold in Taiwan each year to obtain the probability of proportion (%) of imported beef from the USA out of the total amount of fresh beef sold in Taiwan. We also show the probability of proportion (%) of imported beef in Fig 1. This result will use in equation 4 to establish the probability distribution of ingesting E. coli O157:H7 through beef products among the Taiwanese population for each sex and age group (Log10 CFU).

The distribution of E. coli O157 :H7 per serving before cooking obtain from Ccont N (L110), and use in equation 2 to obtain the residual amount of E. coli O157:H7 (CFU/g) in the beef after cooking under specific temperature and time conditions. Although we don't show the distribution of E. coli O157 :H7 per serving before cooking, we have shown the probability distribution of E. coli O157:H7 contamination in beef imported from the USA after cooking (rare, medium, medium–well) in Fig 2.

The rate of contamination (Ccont R) with E. coli O157:H7 will use in equation 3 to obtain the distribution of E. coli O157:H7 in beef after cooking. We assume the distribution rate of contamination of E. coli O157:H7 in beef after cooking the same with before cooking.

For easy reading and know, the related symbols or equation we also highlight the same color in Fig 1.

Cj and the notion of food cluster is not sufficiently described. I read your answer and I think that gender and age should be mentioned in the manuscript. What is a food group ? If there is a distinction between consumption profile, on which basis is performed this distinction ? Why is this distribution applied to already cooked products and not uncooked ?

Answer:
Thanks for your suggestion. We think gender and age should be mentioned in the equation and manuscript.

Consumption profile was obtained from 24-hour dietary recall and food frequency questionnaires of NAHSIT, specific details about how to conduct surveys were presented in Pan (2011).All four surveys were conducted from 2005 to 2008 (children aged below 6 years old and above 19 years), 2010 to 2011 (junior high school students), 2011 (senior high school students), and 2012 (primary school pupils) with different target populations.(L118)

About food group, our beef daily intake data mapping from individual ID and their food recode, ex: ribeye, top cap, beef, steak, and sirloin...we will pool into a beef group. And individual ID consumption could be divided into sex and age groups show in table 1.

Cooked products and uncooked products could use the food expansion ratio to calculate the weight. However, the present study assumed that changes in raw beef and cooked steak were not significant in weight. So we only use cooked beef weight in our distribution.

Line 376 : was it not possible to have the distribution of the population who eat beef in Taiwan ? Indeed it is an important limitation of the study.

Answer:
The consumption rate (CR) (g/kg bw/day) distribution data were gathered from the Nutrition and Health Survey in Taiwan (NAHSIT). Beef and its product daily intake data (n=403) were obtained from 24-hour dietary recall and food frequency questionnaires of NAHSIT (n≒351086 food item/code). We could have the distribution of the population who eat beef in Taiwan. But the present study only calculated individuals who ate beef (consumer-only) and excluded a consumption rate = 0. So we revise the word “impossible” to not.

Reviewer 2 Report

The authors have done an excellent job in revising this manuscript. They have adequately caveated their findings and conclusions per the uncertainties inherent in the model. 

Author Response

We thank the reviewer’ valuable comments to improve the quality of this manuscript.

Round 3

Reviewer 1 Report

Thanks for the answers. I am sorry to insist but I think that equations should be revised to make appear the notations that are used to make the calculations. Otherwise, it is unusefull to explain for example what Ccont N corresponds to if it is not used afterwards in equations. I suggest to replace C0 for example by C cont N if I correctly understood and to put C cont R with j index rather than Cj and make such correction everywhere where it is needed.

Author Response

Thank you very much for Reviewer's comment, it's really very useful. We replaced C0 by C cont N and replaced Cj by C cont R with j index in the equations and text (yellow highlight). After revising the equations, the readers will be easy to read and helpful to find the equations notations in the text.
